# New Facet in Viscometry of Charged Associating Polymer Systems in Dilute Solutions

**DOI:** 10.3390/polym15040961

**Published:** 2023-02-15

**Authors:** Anna Gosteva, Alexander S. Gubarev, Olga Dommes, Olga Okatova, Georges M. Pavlov

**Affiliations:** 1Institute of Macromolecular Compounds, Russian Academy of Sciences Bolshoi pr. 31, 199004 Saint Petersburg, Russia; 2Department of Molecular Biophysics and Polymer Physics, Saint Petersburg State University, Universitetskaya nab. 7/9, 199034 Saint Petersburg, Russia

**Keywords:** associated polymers, intrinsic viscosity, Huggins and Kraemer plots, polymer-polymer interactions

## Abstract

The peculiarities of viscosity data treatment for two series of polymer systems exhibiting associative properties: brush-like amphiphilic copolymers—charged alkylated N-methyl-N-vinylacetamide and N-methyl-N-vinylamine copolymer (MVAA-*co*-MVAC_n_H_2n+1_) and charged chains of sodium polystyrene-4-sulfonate (PSSNa) in large-scale molecular masses (MM) and in extreme-scale of the ionic strength of solutions were considered in this study. The interest in amphiphilic macromolecular systems is explained by the fact that they are considered as micellar-forming structures in aqueous solutions, and these structures are able to carry hydrophobic biologically active compounds. In the case of appearing the hydrophobic interactions, attention was paid to discussing convenient ways to extract the correct value of intrinsic viscosity $\left[\eta \right]$ from the combined analysis of Kraemer and Huggins plots, which were considered as twin plots. Systems and situations were demonstrated where intrachain hydrophobic interactions occurred. The obtained data were discussed in terms of $\ln\eta_{r}$ vs. $c\left[\eta \right]$ plots as well as in terms of normalized scaling relationships where $\eta_{r}$ was the relative viscosity of the polymer solution. The first plot allowed for the detection and calibration of hydrophobic interactions in polymer chains, while the second plot allowed for the monitoring of the change in the size of charged chains depending on the ionic strength of solutions.

## 1. Introduction

Amphiphilic polymers have unique properties that are provided by the association of hydrophobic groups and their micro segregation with hydrophilic fragments in aqueous solutions. The association of such polymers in solutions has been studied extensively [1,2,3,4,5]. Amphiphilic polymers are synthesized in the form of various types of block copolymers or molecular brushes when hydrophobic side chains are attached to the hydrophilic main chain of a polymer. Such macromolecules often contain charged groups. In solutions, they exhibit associative properties, which open up possibilities for creating structures that respond to various stimuli and exhibit reversible properties. With an increase in the concentration of solutions, such polymers generate a network structure, where, due to non-covalent interactions, a system of points of non-covalent bonds between different chains (stickers) is formed, which allow for a labile material. Due to the presence of non-covalent interactions (hydrogen bonds, host–guest interactions, electrostatic interactions), amphiphilic polymers acquire new functions that significantly expand their applications.

Such amphiphilic polymers are based on synthetic structures [6,7,8,9,10,11], as well as on natural polymers [12,13,14,15]. They can be used for enzyme immobilization, controlled drug release [16,17,18], microencapsulation, production of membranes for separation processes, viscosity modifiers for oil production [19,20,21], and as emulsifiers, dispersants, foaming agents, thickeners, etc. The structures formed in solutions of associating polymers depend on the concentration of the polymer, the number of attracting groups in the chains, the strength of the physical bonds they form, and the degree of affinity between the polymer and the solvent. In dilute solutions of amphiphilic polymers, intramolecular self-association (i.e., the formation of physical bonds between attracting groups of the same chain) can occur which can lead to a decrease in the size of an individual macromolecule and the formation of loops in it or the formation of a uni-macro-molecular micelle. At high concentrations, the aggregation of several macromolecules into an intermolecular micelle and then to the formation of a physical gel of polymer chains can be observed. Interchain aggregation in a physical network leads to a sharp increase in the viscosity of the solution and the appearance of its high elasticity. These complex dynamic processes in the solutions of associating polymers are extensively studied by suggesting theoretical models of the system of interest [22,23,24,25,26,27] and rheological methods [28,29,30,31,32].

Rheology studies both the linear viscoelastic behavior of associative polymers, taking into account the density of stickers and the strength of associations with stickers, and the dynamics of associative polymers, determined by the structure of sticker clusters/aggregates and their dissociation [27,33].

In dilute polymer solutions, viscometry serves to determine the characteristics of an isolated macromolecule, following Staudinger [34]. The macromolecules of amphiphilic polymers manifest their peculiar behavior already in dilute solutions, and, as a rule, demonstrate high values of the Huggins parameter [35,36,37,38].

The study of the viscous flow of dilute solutions of natural and synthetic polymers has been one of the cornerstones in establishing their macromolecular nature and today is a rather routine but indispensable means to start any polymer investigation. The intrinsic viscosity $\left[\eta \right]$ is one of the most important and easily accessible quantities that characterizes the size and conformation of linear polymer molecules. The physical meaning of the $\left[\eta \right]$ value of linear polymer molecules is revealed by the Flory–Fox relation: $\left[\eta \right]=\Phi {\langle h^{2}\rangle }^{3/2}/M$, where $\langle h^{2}\rangle$ is the mean square distance between the ends of the macromolecular chain, Φ is the hydrodynamic Flory parameter [39].

The intrinsic viscosity was defined by Staudinger as $\underset{c\to 0}{\lim}\left(\eta_{\mathrm{sp}}/c\right)\equiv \left[\eta \right]$ [34] and by Kraemer as $\underset{c\to 0}{\lim}\left(\ln\eta_{r}/c\right)\equiv \left[\eta \right]$ [40], where $\eta_{r}$ is relative viscosity, $\eta_{\mathrm{sp}}\equiv \eta_{r}-1$ is specific viscosity, and c is the concentration of the dissolved substance in g/cm^3^. This definition emphasizes that the $\left[\eta \right]$ value characterizes a friction of the isolated macromolecule surrounded by solvent molecules. The method of extrapolating the $\left(\eta_{\mathrm{sp}}/c\right)$ and $\left(\ln\eta_{r}/c\right)$ values to c→0 was not indicated by the authors of [34,40]. Various extrapolation plots have been proposed and are still being proposed to determine the intrinsic viscosity of polymers [41,42,43,44,45,46].

For a long time, and even now, in polymer science the Huggins [47] plot is overwhelmingly used to determine the $\left[\eta \right]$ of polymer molecules:(1)$$\eta_{\mathrm{sp}}/c={\left[\eta \right]}^{H}+k^{\prime }{\left[\eta \right]}^{2}c+\ldots .$$

Sometimes the Mead–Fuoss plot [46,48] is used. It is also known as the Kraemer plot [40]:(2)$$\ln\eta_{r}/c={\left[\eta \right]}^{K}+k^{\prime \prime }{\left[\eta \right]}^{2}c+\ldots .$$

Intrinsic viscosity values determined by Equations (1) and (2) are indicated in the text as ${\left[\eta \right]}^{H}$ and ${\left[\eta \right]}^{K}$, respectively. At c→0, these plots should lead to the equivalent estimates of $\left[\eta \right]$ value because $\ln\eta_{r}=\ln\left(1+\eta_{\mathrm{sp}}\right)$ can be expanded into a sign-alternating series in $\eta_{\mathrm{sp}}$ when $\eta_{\mathrm{sp}}\leq 1$. The equality of the values ${\left[\eta \right]}^{H}={\left[\eta \right]}^{K}$ is an axiomatic condition, i.e., must be performed with a highest level of accuracy. In addition, the mathematical result of this expansion into a mathematical series is: $k^{\prime \prime }\equiv k^{\prime }-0.5$. As Garcia de la Torre et al. aptly noted [49], the last correlation is just a mathematical consequence that does not correlate with the chemical and physical features of polymer solutions, i.e., the validity of the latter relationship does not depend on either the nature of the solvent or polymer–solvent interactions. Note that the ratio $k^{\prime \prime }\equiv k^{\prime }-0.5$ is not fulfilled in many cases. Unfortunately, in the literature there are only few works with the simultaneous presence of the two above-mentioned methods ((1) and (2)). In rare cases, the authors only mention the Kraemer plot, as a rule, without comparing the results obtained. For example, The Polymer Handbook [50] contains extensive tables of the Huggins parameter, but there are no tables for $k^{\prime \prime }$.

In this work, the viscometry data were thoroughly analyzed from a methodological and metrological point of view. We demonstrated the obvious need to use the Huggins–Kraemer twin plot in the study of dilute solutions of amphiphilic polymers, as well as neutral linear polymers in thermodynamically poor solvents. First, we demonstrated the evolution of the twin plot for linear polymers as the solvent quality worsening. Then we analyzed the viscometric behavior of the strong linear polyelectrolyte (sodium polystyrene-4-sulfonate (PSSNa)) over wide ranges of both molecular masses (MM) and ionic strengths. In addition, the behavior of brush-like amphiphilic copolymers strong polyelectrolyte (alkylated copolymer of N-methyl-N-vinylacetamide and N-methyl-N-vinylamine (MVAA-co-MVAC_n_H_2n+1_)) with hydrophobic groups of different lengths in wide ranges of MM and ionic strengths were considered, as well as some data from the literature [51,52]. Finally, the results were discussed in terms of the concept of the normalized Kuhn-Mark-Houwink-Sakurada (KMHS) relationship, which made it possible to compare the results with the full spectrum of possible conformational states of linear macromolecules.

## 2. Materials and Methods

The synthesis of polymer systems under discussion was described in works [53,54,55,56]. The structures of polymers are shown in Figure 1.

The polymers were investigated earlier in water and salt-water solutions by the methods of molecular hydrodynamics: viscometry, sedimentation velocity, and translational diffusion. The results for poly-N-methyl-N-vinylacetamide (PMVA) are described in [57]; for hydroxypropylmethylcellulose (HPMC) in [58]; for PSSNa in [54,59]; for MVAA-*co*-MVAC_n_H_2n+1_ in [55,60,61]. The copolymer bears charges at the junction of the main and each side chain. The composition of the copolymer is 15 mol.% side chains.

All investigations were carried out at 25 °C. The viscous flow of dilute solutions was investigated in Ostwald capillary viscometer with solvent flow times: 83.5 s (H_2_O), 84.1 s (aqueous 0.1 M NaCl), 84.3 s (aqueous 0.2 M NaCl), 120 s (aqueous 4.17 M NaCl). The molecular masses $M_{\mathrm{sD}}$ were determined through the Svedberg relationship $M_{\mathrm{sD}}=R\left[s\right]/\left[D\right]$, where $\left[s\right]=s_{0}\eta_{0}/\left(1-\bar{\upsilon }\rho_{0}\right)$ is the intrinsic sedimentation coefficient, $\left[D\right]=D_{0}\eta_{0}/T$ is the intrinsic diffusion coefficient, and R is the gas constant. Translational diffusion was studied with a Tsvetkov polarizing diffusometer [48] in a cell [62,63] with optical path 30 mm by classical method of forming a boundary between solvent and solution. The velocity sedimentation was investigated with a Proteomelab XLI (Beckman) ultracentrifuge in a 12-mm two-sector cell at a rotor speed of 40,000 rpm. Sedimentation interference curves were processed with the Sedfit software using general scaling law approach [64,65]. The buoyancy factor $\Delta \rho /\Delta c=\left(1-\bar{\upsilon }\rho_{0}\right)$ was measured with a DMA-4000 densitometer (Anton Paar).

## 3. Results and Discussion

### 3.1. From Linear Polymers in Thermodynamically Good Solvent through Marginal One and θ-Solvent to Amphiphilic Graft Copolymers

#### 3.1.1. Hydrophylic Flexible Noncharged Nonassociating Poly-N-Methyl-N-Vinylacetamide (PMVA)

For flexible-chain polymers in thermodynamically good solvents, the $k^{\prime \prime }$ parameter is negative and smaller in modulus than the $k^{\prime }$ parameter. Therefore, the dependence $\ln\eta_{r}/c$ vs. c changes more slowly with concentration than the dependence $\eta_{\mathrm{sp}}/c$ vs. c. For this reason, as Flory noted, extrapolation ${\left(\ln\eta_{r}/c\right)}_{c\to 0}$ is more preferable than ${\left(\eta_{\mathrm{sp}}/c\right)}_{c\to 0}$ [66] (page 325). However, Flory’s note was not accepted by the polymer community, and the vast majority of information published up to date on the determination of the $\left[\eta \right]$ value is derived only from Huggins plots. (See, for example, [50]).

The Huggins and Kraemer classical twin plot for flexible noncharged PMVA in water is shown in Figure 1. In the range of relative viscosities $1.08<\eta_{r}<1.84$ the following results were obtained: from the Huggins plot ${\left[\eta \right]}^{H}=\left(107.4\pm 0.7\right)\;{\mathrm{cm}}^{3}/g,$ $k^{\prime }=0.40\pm 0.02,$ r=0.9966 and from the Kraemer plot ${\left[\eta \right]}^{K}=\left(107.6\pm 0.5\right)\;{\mathrm{cm}}^{3}/g,$ $k^{\prime \prime }=-\left(0.12\pm 0.01\right),$ r=0.9789 (Sample 4, Table 1). These results testify both the equivalence of the assessment of the main quantity—intrinsic viscosity—by Huggins and Kraemer plots and the validity of the relation $k^{\prime \prime }=k^{\prime }-0.5$. The data for other PMVA samples [57] are shown in Table 1.

The average deviation between the ${\left[\eta \right]}^{H}$ and ${\left[\eta \right]}^{K}$ values is 0.7%, and $(k^{\prime }-k^{’’})=0.50\pm 0.015$ over the entire Table 1 data array. It should be noted that for flexible-chain polymers in thermodynamically good solvents, relations (1) and (2) usually lead to virtually the same value $\left[\eta \right]$, and the ratio $k^{\prime \prime }=k^{\prime }-0.5$ is also practically fulfilled. This kind of accuracy is common for the viscosity measurements of flexible macromolecules.

This led to the fact that in the most studies, the use of the Kraemer plot was considered an unnecessary waste of time and the process of determining $\left[\eta \right]$ value has been optimized as much as possible, i.e., only the Huggins plot was used.

#### 3.1.2. Peculiarities of Viscosity Behavior of Associating Systems

As the thermodynamic quality of the solvent worsens, the absolute value of the parameter $k^{\prime \prime }$ decreases to zero, and then becomes positive and continues to increase. The parameter $k^{\prime }$ continuously increases in this case.

The slope of $\ln\eta_{r}/c$ vs. c (Equation (2)) is the quantity $\left(k^{’’}{\left[\eta \right]}^{2}\right)$. When $k^{\prime \prime }\approx 0$, this dependence in the region of dilute solutions looks like some “fluctuation” of $\ln\eta_{r}/c$ values depending on the concentration. Apparently, it is this range of $k^{\prime \prime }$ values that induces doubt among experimenters about the reliability of intrinsic viscosity determining from Kraemer plot. Indeed, in this case, this dependence may have a small or extremely small linear correlation coefficient. However, *this does not mean that in this case the main sought value* $\left[\eta \right]$ *becomes less certain when using the Kraemer plot compared to the Huggins plot*. When $k^{\prime \prime }\approx 0$ the absolute error in determining $k^{\prime \prime }$ is greater than its value. But nevertheless, the absolute error in determining ${\left[\eta \right]}^{K}$ is practically the same as that of ${\left[\eta \right]}^{H}$ for the same data set. The $k^{\prime \prime }\approx 0$ is typical for thermodynamically poor or marginal solvents, i.e., solvents approaching the θ-solvent in terms of their thermodynamic quality.

An example of such a rare system is shown in Figure 2 with a treatment of our viscometric data for a water-soluble hydroxypropylmethylcellulose (HPMC) sample with the degrees of substitution of 1.90 for methoxy and 0.26 for hydroxypropyl groups and with a MM of 207,000 g/mol. MM was calculated from the KMHS ratio established for HPMC samples studied in water [58].

The difference between the estimates of $\left[\eta \right]$ from the two plots (Figure 2a) is ${\left[\eta \right]}^{K}/{\left[\eta \right]}^{H}\approx 2.5\%$, which is usually not considered significant. Obviously, in this case, there is a good agreement between the ${\left[\eta \right]}^{H}$ and ${\left[\eta \right]}^{K}$ values. However, this is 3.5 times greater than the deviation observed for the flexible chain PMVA (Table 1). It is noteworthy that the coefficient of linear correlation of the Huggins plot is r=0.9984, whereas this coefficient for the Kraemer plot is r=0.0637, which means that the slope of the Kraemer plot is practically zero. At the same time, the absolute errors of the $\left[\eta \right]$ value for the two plots practically coincide $\Delta {\left[\eta \right]}^{K}=\Delta {\left[\eta \right]}^{H}=4\;{\mathrm{cm}}^{3}/g$.

The value of the Kraemer parameter $k^{\prime \prime }\approx 0$ indicates that water is not a thermodynamically good solvent for hydroxypropylmethylcellulose. It can be expected that as the concentration of solutions increases, the dependences will clearly show the hydrophobicity of this polymer. Indeed, Figure 2b shows a noticeable difference in the behavior of the Kraemer and Huggins dependences in the interval $c\left[\eta \right]>1$. The dependence of $\eta_{\mathrm{sp}}/c$ on c increases sharply and can be approximated by a parabola, while the dependence of $\ln\eta_{r}/c$ changes slightly. In this regard, we have carried out a correction of the $\eta_{\mathrm{sp}}/c$ in the low concentrations. For this, we formed a data system consisting of experimental values $\eta_{\mathrm{sp}}/c$ at the $\eta_{r}<1.4$ and add to them a point (c=0, $\eta_{\mathrm{sp}}/c={\left[\eta \right]}^{K}$), taking into account an axiom ${\left[\eta \right]}^{H}\equiv {\left[\eta \right]}^{K}$. We processed this data with a second degree polynomial function and draw the tangent to it at c=0. The tangent equation is $\;\eta_{\mathrm{sp}}/c=540+10.25c$, i.e., ${\left[\eta \right]}_{\mathrm{corr}}^{H}=540$, $k_{\mathrm{corr}}^{’}=0.35$. Thus, a slight increase in ${\left[\eta \right]}^{H}$ by 2.3% leads to a twofold decrease in the Huggins parameter. In a similar way, we processed the data of $\eta_{\mathrm{sp}}/c$ in the region of extremely dilute solutions for the amphiphilic polymer systems.

Note that Garcia de la Torre et al. had already developed the program for linear least-square fits with a common intercept [67] for systems with a negative value of the Kraemer parameter. It would be useful to adapt the program for joint processing of the Huggins and Kraemer plots for the associating polymer systems with a positive value of the Kraemer parameter, as suggested above and below.

With a further decrease in the polymer-solvent affinity and an increase in the role of polymer-polymer interactions compared to polymer-solvent interactions, the parameter $k^{\prime \prime }$ changes sign to positive. The relation $k^{\prime \prime }=k^{\prime }-0.5$ is no more valid. However, in all cases, the $k^{\prime \prime }$ parameter is less in absolute value than the parameter $k^{\prime }$, therefore, the dependence $\ln\eta_{r}/c$ will remain linear in a larger concentration range compared to the dependence $\eta_{\mathrm{sp}}/c$.

This situation is illustrated by an charged brush-like alkylated amphiphilic copolymer of N-methyl-N-vinylacetamide and N-methyl-N-vinylamine, which combines a hydrophilic charged base and hydrophobic C_12_H_25_ side groups (MVAA-*co*-MVAC_12_H_25_) [55,61] (Figure 3) and linear homopolymer PSSNa, which exhibits hydrophobic interactions in θ-solvent [54,59,68] (Figure 4).

The results obtained in this way for all investigated PSSNa fractions in aqueous 4.17 M NaCl, are presented in Table 2. The ${\left[\eta \right]}^{H}$ values obtained by direct linear extrapolation of $\eta_{\mathrm{sp}}/c$ (let’s indicate it ${\left[\eta \right]}_{\mathrm{brutto}}^{H}$) in the entire dataset are lower than ${\left[\eta \right]}^{K}$ values, and the error in determining ${\left[\eta \right]}^{H}$ is noticeably greater than that for ${\left[\eta \right]}^{K}$. After the $\left[\eta \right]$ values are converged to a single value corresponding to ${\left[\eta \right]}^{K}$, the corrected values of the Huggins parameter are reduced by 2–6 times compared to the $k_{\mathrm{brutto}}^{’}$ values obtained by simple linear treatment (Figure 3a, line 1, Figure 4a, line 1). A consistent estimate of $\left[\eta \right]$ will lead to a change in both parameters of the Kuhn-Mark-Houwink-Sakurada ratio and, accordingly, to a refinement of the conformational status of PSSNa macromolecules in aqueous 4.17 M NaCl compared to estimates based on ${\left[\eta \right]}^{H}$ [54].

The most important postulate when using the Huggins and Kraemer plots is the statement that $\underset{c\to 0}{\lim}\left(\eta_{\mathrm{sp}}/c\right)\equiv \underset{c\to 0}{\lim}\left(\ln\eta_{r}/c\right)$, i.e., the dependencies must converge at one point on the y-axis at c=0. However, for many polymer systems the intercept on the y-axis at c=0 on Huggins plot is less than that on Kraemer one. That is, from the Huggins plot, we get a somewhat underestimated value of the intrinsic viscosity, while the brutto values of the Huggins parameter will be quite large, the larger is the value of $k^{\prime \prime }$. In the considered Figure 2, Figure 3 and Figure 4, the initial values of ${\left[\eta \right]}_{\mathrm{brutto}}^{H}$ are less than <!-- MathType@Translator@5@5@MathML2 (no namespace).tdl@MathML 2.0 (no namespace)@ -->

${\left[\eta \right]}_{\mathrm{brutto}}^{H}$ by 2.3, 7.1 and 34%, respectively. When ${\left[\eta \right]}^{H}$ is corrected to ${\left[\eta \right]}^{K}$, the Huggins parameter decreases by factors of 2.0, 1.7, and 2.8, respectively, compared to its brutto values. The correlation between $k^{\prime }$ and $k^{\prime \prime }$ over the entire range of their values requires more detailed study for a wide set of polymer systems.

### 3.2. Viscometric Data at Different Ionic Strengths of Solutions

#### 3.2.1. Sodium Poly(Styrene-4-Sulfonate)

Equation (2) can be represented as the dependence of $\ln\eta_{r}$ on the Debye parameter ($c\left[\eta \right]$):(3)$$\ln\eta_{r}=c\left[\eta \right]+k^{\prime \prime }{\left(c\left[\eta \right]\right)}^{2}+\ldots .$$

The Debye parameter $c\left[\eta \right]$ characterizes the dilution degree of the solution.

This is a well-known plot in rheology which allows one to compare viscosity data for polymers with various molecular masses and chemical structures, and in different solvents.

Over a wide concentration range the dependence $\ln\eta_{r}$ on $c\left[\eta \right]$ is described by a second-degree polynomial:(4)$$\ln\eta_{r}=A+B_{1}\left(c\left[\eta \right]\right)+B_{2}{\left(c\left[\eta \right]\right)}^{2}+\ldots .$$

According to the Kraemer definition of $\left[\eta \right]$, its initial slope $B_{1}$ (at c→0) is equal to 1. The second derivative $B_{2}$, determined in the region of dilute solutions, is the Kraemer parameter $k^{\prime \prime }$. Its sign determines the trend of the dependence. Thus, at c→0 with A≈0, $B_{1}\approx 1$, and $B_{2}\approx k^{\prime \prime }$ the relationship (4) is transformed into (3).

The plot $\ln\eta_{r}=f\left(c\left[\eta \right]\right)$ for PSSNa in aqueous 4.17 M NaCl (a), aqueous 0.2 M NaCl (b), and in salt-free water (e) solutions at 25 °C as well as the data for two polystyrene (PS) samples with MM 146,000 and 600,000 g/mol in toluene at 30 °C (points c and d) [52] are shown in Figure 5.

The points for PSSNa in aqueous 0.2 M NaCl solutions (b in Figure 5) for all fractions fit into virtualy single convex dependence, i.e., characterized by a negative second derivative ($B_{2}<0$), which corresponds to a negative value of the Kraemer parameter (Table 3). Note that the data on the toluene-soluble fractions of PS (c, d in Figure 5) [52], fit well into a single dependence with PSSNa in aqueous 0.2 M NaCl. This means that these macromolecules are in virtually identical thermodynamic conditions. The PSSNa data in aqueous 4.17 M NaCl solution (a θ-solvent at 25 °C [54,69], form a single concave curve, which corresponds to positive values of both $B_{2}$ and $k^{\prime \prime }$. The space defined by coordinates $\ln\eta_{r}=f\left(c\left[\eta \right]\right)$ can be divided into two parts by a straight line $\ln\eta_{r}=c\left[\eta \right]$ (bisecrix) corresponding to the condition $B_{2}=0$ ($k^{\prime \prime }=0$). To the right is an area of hydrophilic systems and/or systems that are in thermodynamically good conditions; to the left is an area of hydrophobic and/or systems near θ-conditions. Note that, both in aqueous 0.2 M and 4.17 M NaCl solutions, the system of PSSNa fractions manifests itself as a molecular homologues system. In salt-free solutions (points e in Figure 5), the fractions of charged PSSNa macromolecules do not form a single dependence. This situation will be discussed below.

In contrast with data in Table 2 (PSSNa in 4.17 M NaCl) the average deviation between the ${\left[\eta \right]}^{H}$ and ${\left[\eta \right]}^{K}$ values is 0.8%, and $\left(k^{\prime }-k^{\prime \prime }\right)=0.48\pm 0.01$ over the entire Table 3 data array in 0.2 M NaCl. This is similar to statistic of Table 1 for noncharged PMVA in H_2_O.

Note that despite the demonstration of associative properties in 4.17 M NaCl, PSSNa macromolecules behave both in 0.2 M and 4.17 M NaCl as a homologous series (Figure 5).

#### 3.2.2. Alkylated Random Copolymers of N-Methyl-N-Vinylacetamide and N-Methyl-N-Vinylamine (MVAA-*co*-MVAC_n_H_2n+1_)

For brush-like amphiphilic copolymers with hydrophobic side chains and hydrophilic backbone an important structural parameter is the ratio of the average distance between two neighboring alkyl side chains ($l_{\mathrm{phil}}$) to contour length of their side chain ($l_{\mathrm{phob}}$). This parameter ($l_{\mathrm{phil}}/l_{\mathrm{phob}}$) characterizes the ratio between hydrophilic and hydrophobic parts in a copolymer molecule. The MVAA-*co*-MVAC_12_H;_25_ random copolymer has such a composition of the side aliphatic groups (15 mol.%) when the average distance between adjacent side groups is almost equal to the contour length of the C_12_H_25_ side group ($l_{\mathrm{phil}}\approx l_{\mathrm{phob}}$). These copolymers, being strong polyelectrolytes, are readily soluble in salt-free water, and being flexible-chain in nature, they strongly react to changes in the ionic strength of the solution.

As follows from the viscometric plots, the primary polyelectrolyte effects are suppressed in aqueous 0.1 M NaCl solutions. The viscometric data for MVAA-*co*-MVAC_n_H_2n+1_ are given in Figure 6 and Table 4. The data in Figure 6 are presented in $\ln\eta_{r}$ vs. $c\left[\eta \right]$ coordinates with the division of space into two regions: hydrophobic and hydrophilic.

Despite the fact that the samples have the same chemical composition, the viscometric results do not fit into a single curve but represent a set of curves with different values of the positive second derivative $B_{2}$. Thus, the samples can be differentiated by the level of hydrophobicity: the more $B_{2}$, the more the sample is hydrophobic. As a result, the homology of the studied series is violated, which is associated evidently with their irregular, random distribution of C_12_H_25_ side groups along the chain.

In DMF + 0.1 M LiCl solutions of MVAA-*co*-MVAC_12_H;_25_ the hydrophobicity effect “turns off” and the viscometric results (points 6–8, Figure 6) are in good agreement with the data obtained for the parent uncharged homopolymer PMVA (dashed curve, Figure 6). In organic solvents, where hydrophobic interactions are absent, these copolymers behave as polymer homologues.

Note that samples with short aliphatic groups C_6_H_13_ and C_8_H_17_ (points 9 and 10, Figure 6) do not exhibit hydrophobic behavior in 0.1 M NaCl solutions at all, being located near the data corresponding to the homopolymer. For these copolymers (with the composition mol. 15%), the contour length of the side radicals is much less than the average distance between them along the main chain. In this case, the total share of hydrophobicity of the macromolecule is less than the share of hydrophilicity.

For the studied copolymers the ratio $l_{\mathrm{phil}}/l_{\mathrm{phob}}$ changes within the limits: $1.03<l_{\mathrm{phil}}/l_{\mathrm{phob}}<1.90$. When $l_{\mathrm{phil}}/l_{\mathrm{phob}}\approx 1$, intramolecular hydrophobic interactions occur in the copolymer; when $l_{\mathrm{phil}}/l_{\mathrm{phob}}>1.5$ they are not observed. The latter is shown in Figure 6 for MVAA-*co*-MVAC_6_H_13_ (points 9) and MVAA-*co*-MVAC_8_H_17_ (points 10) in aqueous 0.1 M NaCl solutions.

#### 3.2.3. The Behaviour of Polymers in Salt-Free Solutions

The two studied systems PSSNa and MVAA-*co*-MVAC_12_H_25_ are strong polyelectrolytes. The $\left[\eta \right]$ value was determined from initial slope of the dependence of $\ln\eta_{r}$ on c according to the Kraemer definition $\underset{c\to 0}{\lim}\left(\ln\eta_{r}/c\right)\equiv \left[\eta \right]$ [40,56,70].

The results on PSSNa are presented at Figure 7 and Figure 8, and in Table 5. For comparison, the data from the unique viscometric study of PSSNa in H_2_O [51] were added to Figure 8.

In salt-free solutions of highly charged linear chains, electrostatic interactions are dominant.

Irregular re-condensation of counterions onto a charged chain leads to a statistical distribution of charges along the chain and loss of homology of such polymer series in salt-free solutions.

One can see that the points for all charged polymers (a, b, c in Figure 8) lie below the line d for a neutral water-soluble homopolymer PMVA. It is explained by the fact that with an increase in the concentration of the polyelectrolyte in pure water, the ionic strength of the solution increases due to an increase in the number of counterions in the solution, and the coils shrink. As *c*[*η*] increases, the interaction of hydrophobic C_12_ side groups slightly appears. The dependence at Figure 8 demonstrates a weak manifestation of the hydrophobicity of MVAA-*co*-MVAC_12_H_25_ in comparison with the PSSNa. The MVAA-*co*-MVAC_12_H_25_ data (points c in Figure 8) lie a little bit lower in the region $c\left[\eta \right]>2$ than that for PSSNa (points a, b in Figure 8).

At very high values of polymer charge density and low ionic strength, the level of hydrophobicity practically does not affect the size of polyelectrolyte chains.

### 3.3. Normalized Scaling Relationships: Comparison of Viscometric Data

Intrinsic viscosity is the most sensitive characteristic to the size, shape, and their change for linear macromolecules in solutions. This is why the Kuhn-Mark-Houwink-Sakurada relationship ($\left[\eta \right]=K_{\eta }M^{b_{\eta }}$) is so popular in polymer science. To date, an extensive library of these relationships for polymers of various structures and in various solvents has been formed and is being replenished [71]. If we put all these data on the $\mathrm{lg}\left[\eta \right]$ vs. lgM plot, we get a rather chaotic filling of this two-dimensional space. However, it turned out that this “pointillist picture” is transformed into a fairly clear ordered structure of the viscometric spectrum of linear macromolecules of different nature and structure, unfolded in size, if we take into account on this plot the mass of a linear chain length unit ($M_{L}$), which characterizes the linear density of the macromolecule [72,73,74]. This transformation follows from the fundamental Flory–Fox relation that $\left[\eta \right]M_{L}\sim {<h^{2}>}^{3/2}/L\sim V/L$, where V is the volume occupied by the macromolecule in solution. Consequently, the value $\left[\eta \right]M_{L}$ characterizes the volume occupied by the chain segment corresponding to the unit contour length of the macromolecule (which is conveniently chosen as a repeating link). This value will be the greater, the greater is the equilibrium rigidity of the macromolecule and the better is the thermodynamic quality of the solvent. Thus, by transforming the standard plot $\mathrm{lg}\left[\eta \right]=f\left(\mathrm{lg}M\right)$ used to establish the classical KMHS relation, we obtained the plot ($\mathrm{lg}\left(\left[\eta \right]M_{L}\right)=f\left(\mathrm{lg}\left(M/M_{L}\right)\right)$, which represents a sweep by their sizes of the entire spectrum of linear macromolecules so far studied. Linear macromolecules were divided into the following classes: extra-rigid, rigid, flexible in thermodynamically good solvents, flexible under θ-conditions, and globular. This plot is the result of an analysis of a large number of data in the literature and covers the area of macromolecules contour lengths changing by three orders of magnitude and the area of equilibrium rigidities of linear chains changing by approximately 500 times. Thus, by applying the experimental data to the obtained template/sweep, one can judge whether the studied polymer belongs to one or another class of linear polymers and get an idea of the length of the statistical segment or the persistent length of the chain. Let us analyze the results discussed in this paper in the coordinates $\mathrm{lg}\left(\left[\eta \right]M_{L}\right)$ vs. $\mathrm{lg}\left(M/M_{L}\right)$ (Figure 9).

The joint manifestation of electrostatic long-range effects and short-range interaction leads to the maximum sizes of charged chains in salt-free solutions of both PSSNa chains and MVAA-*co*-MVAC_12_H_25_ chains (points 2 and 5). In the salt-free solutions, the hydrophobicity of the chains does not manifest in any way. Despite the different composition (PSSNa—100 mol.% and MVAA-*co*-MVAC_n_H_2n+1_—15 mol.%), these chains are strong polyelectrolytes [61,75]. As the ionic strength of the solutions is increased to 0.1–0.2 M NaCl, the polyelectrolyte effects are largely suppressed, as indicated by the linear viscometric dependences (Equations 1 and 2). The data system shifts to the region of flexible chain macromolecules (points 3 and 6). Here, in MVAA-*co*-MVAC_12_H_25_ chains, hydrophobic interactions begin to manifest to a greater extent than in linear PSSNa chains. As the ionic strength increases the MVAA-*co*-MVAC_12_H_25_ [η] value decreases insignificantly (points 7, 8, Figure 9). With a further increase in the ionic strength of solutions, MVAA-*co*-MVAC_12_H_25_ samples cease to dissolve molecularly. The mobility of -C_12_H_25_ side groups and their interaction in aqueous solutions are facilitated in comparison with the interactions of the elements of the same linear chain, as occurs in PSSNa chains. This is due to the fact that the side chains of graft copolymer chains are richer in entropy than linear chains of PSSNa. It is the copolymer side chains that cause hydrophobic interactions in these copolymers. Therefore, the manifestation of nonmolecular solutions and the onset of gelation occur earlier for MVAA-*co*-MVAC_12_H_25_ copolymers than for PSSNa.

Under the θ-condition (4.17 M NaCl, 25 °C), the PSSNa chains are additionally compressed (points 4) and their sizes fall between the θ-conditions for conventional flexible-chain polymers and the globular state. Under these conditions, the standard relation for PSSNa chains has the form: ${\left[\eta \right]}_{4.17\;M}=0.0464M^{0.43\pm 0.04}$, where the scaling index $b_{\eta }<0.5$ that is, it becomes less than the minimum value allowed for linear macromolecules. Thus, the macromolecules of the strong polyelectrolyte PSSNa, going from top to the bottom, change their conformation from a slightly bent rod to a coil, and then approach the globular conformation, practically realizing the entire conformational spectrum of linear chain macromolecules.

## 4. Conclusions

The study of the viscous flow of dilute solutions of associating polymers allows to detect the presence of hydrophobic interactions in solutions already in the range of their sufficient dilution at $c\left[\eta \right]<0.5$. Obtaining such information will require the comparative use of both the Huggins plot ($\eta_{\mathrm{sp}}/c$ on c) and the Kraemer plot ($\ln\eta_{r}/c$ on c), in a method we call the twin plot. It should be noted that this technique was quite well known in the early 50 s of the 20th century, but then forgotten. Indeed, new is well-forgotten old!

When interpreting the results of studying the viscous flow of polymer solutions, it should be taken into account that both $\underset{c\to 0}{\lim}\left(\eta_{\mathrm{sp}}/c\right)$ and $\underset{c\to 0}{\lim}\left(\ln\eta_{r}/c\right)$ should converge at the same point. In this case, the concentration intervals used for linear extrapolation of the $\eta_{\mathrm{sp}}/c$ and $\ln\eta_{r}/c$ to c→0 can/should be different. Since always $|k^{’’}|<k^{\prime }$, the interval used for extrapolation of $\eta_{\mathrm{sp}}/c$ to c→0 can be much shorter than that for $\ln\eta_{r}/c$ dependence. Recall that the *only* results/part of the results are obtained in the region of dilute polymer solutions when $c\left[\eta \right]<1$ and $\eta_{r}<2$, are the subject to analysis. The final measurement results must contain a single intrinsic viscosity value and correctly determined Kraemer and, especially, Huggins parameters. The values of the Huggins parameter exceeding 1, given anywhere in the literature, are overestimated. In turn, this leads to some underestimated $\left[\eta \right]$ values.

The parameter $k^{\prime \prime }$ is a sign-changing parameter, changing its sign from negative to positive for polymer systems exhibiting significant polymer–polymer interactions in solutions. Such systems include polymers in θ-solvents and amphiphilic copolymers of various topologies. Thus, the parameter $k^{\prime \prime }$ is a source of additional information about the manifestation of polymer–polymer interactions in solutions of macromolecular compounds.

The plot $\ln\eta_{r}$ vs. $c\left[\eta \right]$ is very informative when comparing homologous series of different kind of polymers or the same series in different solution conditions. This plot allows the series under study to be separated into hydrophilic or hydrophobic systems, depending on the sign of the curvature of the second-degree polynomial describing evolution of $\ln\eta_{r}$ in function of the degree of dilution. A positive sign corresponds to hydrophobic systems, and a negative one—to hydrophilic ones. A similar separation will occur between polymers in thermodynamically good solvents and polymer systems near θ-conditions.

A special situation arose in the study of amphiphilic polyelectrolytes in salt-free solutions. The results of the viscous flow of such solutions demonstrated the breach of homology, which also sometimes manifested under suppressed polyelectrolyte effects (compare sodium polystyrene-4-sulfonate and alkylated statistical copolymer N-methyl-N-vinyl acetamide and N-methyl-N-vinyl amine, Figure 5 and Figure 6). Apparently, this can be associated with the irregular re-condensation of some of the counterions on the charged chain, which will lead to a random distribution of the charges remaining on the chain. This means a loss of homology and appears in the irregularity of ln *η*_r_ vs. *c*[*η*] dependence. An informative and powerful tool in interpreting the molecular dependence of intrinsic viscosity is the use of the Kuhn-Mark-Houwink-Sakurada plot normalized to the mass per unit length of a linear polymer chain. Using the pattern established on the basis of the analysis of the literature data bank, one can semi-quantitatively establish the belonging of the studied polymer chains to one or another class of polymers. These possibilities were demonstrated on two series of polymers studied in various solvents.

## Data Availability

Data is contained within the article.

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
