# Peer review of "New Facet in Viscometry of Charged Associating Polymer Systems in Dilute Solutions"

_polymers, 2023, doi:10.3390/polym15040961_

Round 1
Reviewer 1 Report
The present paper is a valuable contribution to the field of polymer characterization in solutions. It comprises two relevant topics (1) the determination of the intrinsic viscosity, [eta], and (2) the significance of [eta] for the study of the polymer conformation and behavior in solution. The work is focused on polymer systems which present either association or polyelectrolyte effects. The first topic is particularly important, because the evaluation of [eta] presents peculiarities in those systems. The importance of doing twin plots, particularly for non-conventional polymer is well justified. New ideas are presented for the treatment of experimental data. I therefore consider that this paper quite interesting and merits publication.
I just have some comments of minor importance.
Title and Introduction : “viscometry of charged associating polymer systems”. Later, when presenting the results for the various systems, it should be clearly stated whether the system has associating and/or electrostatic polyelectrolyte interactions. Is PSSNa in NaCl an associating system.
pp.2,3,5 The authors properly affirm: “...the obvious need to use the Huggins -- Kraemer twin-plot in the study of dilute solutions of amphiphilic polymers, as well as neutral linear polymers in thermodynamically poor solvents”. There is a method yielding an unique [eta].; please see "Multiple Linear Least-Squares Fits with a Common Intercept. Application to the Determination of Intrinsic Viscosity of Macromolecules in Solution". Journal of Chemical Education 80, 1036-1038 (2003). Perhaps this may lead to a somehow better estimations of k’ and k”. It may be worth to try this procedure in a few cases. Indeed, one may note the deviations of k’ among samples over the average (p. 5, Table 1) . I understand that the +- errors are uncertainties in the linear fits. I wonder what the errors would be if expressed as deviations over results of successive full/independent experiments. The change from 0.44+-0.21 for sample 9 vs. 0.05+-0.03 for sample 11 is noticeable (big differences in value and error). If sample 9 were considered an outlier and suppressed in Table 1, the resulting average, about 0.50 would not be much affected.
p. 4-5-6 The finding that for a conventional non-associating, non-electrolyte polymer in good solvents, k’-k” = 0.50+-0.015 (+-3%) with such small variation for over 12 samples is really rewarding. Good!! However, in section 3.1.2, the results for hydrophobic HPMC, k’-k” = 0.70 and k’-k” = 1.22 for the amphiphilic MV copolymer are both for only one viscometric experiment (just one sample)
In the second part of the paper, the authors present quite original treatments of the experimental data. The one based on ln eta_r vs. c[eta] (section 2.2). is quite simple and effective- Fig. 5 provides a nice classification of the behaviors of different (solvent affinity) of the polymer systems .As B_2=k’’, it would be wise to give the B_2 values for the parabolic curves in the legends of Figures 5 and 6 or in the text. The other treatment based on the lg([eta]M_L vs. lg L plot (section 2.3), proposed by Pavlov’s group in refs. 71 and 72. It is a bit speculative but seemingly correct,, and deserves to be considered by workers in this field. I just don't see the values (or references) for the M_L of the various polymers.
Finally, some text editing:
p. 2 “As de la Torre et al…” Please, write “As Garcia de la Torre et al…”
p. 5, Figure 1, legend of Y axis, I think ln eta_r should be ln eta_r/c
p. 10 Figure 5: broken text: [S. G. Weissberg,]
p. 4 “The polymers were… Although it is later clear, please say right here that they are in water or NaCl aqueous solutuon
p. 4 “…as Flory noted, extrapolation (lnr/c)c→0 is more preferable… Please cite the Flory’s publication were this was affirmed. Is it ref. 39 or ref. 66 ?
Author Response
Dear Reviewer,
Thank you very much for your careful reading of our work and useful comments and considerations. Below we answer to all points mentioned. All revisions made to the manuscript we marked up using the“Track Changes” function
Here are the answers to all questions raised:
- Title and Introduction: “viscometry of charged associating polymer systems”. Later, when presenting the results for the various systems, it should be clearly stated whether the system has associating and/or electrostatic polyelectrolyte interactions. Is PSSNa in NaCl an associating system.
Answer: Now the text included clarifications on all presented polymer systems.
- 2,3,5 The authors properly affirm: “...the obvious need to use the Huggins - Kraemer twin-plot in the study of dilute solutions of amphiphilic polymers, as well as neutral linear polymers in thermodynamically poor solvents”. There is a method yielding an unique [eta].; please see "Multiple Linear Least-Squares Fits with a Common Intercept. Application to the Determination of Intrinsic Viscosity of Macromolecules in Solution". Journal of Chemical Education 80, 1036-1038 (2003). Perhaps this may lead to a somehow better estimations of k’ and k”. It may be worth to try this procedure in a few cases. Indeed, one may note the deviations of k’ among samples over the average (p. 5, Table 1). I understand that the +- errors are uncertainties in the linear fits. I wonder what the errors would be if expressed as deviations over results of successive full/independent experiments. The change from 0.44+-0.21 for sample 9 vs. 0.05+-0.03 for sample 11 is noticeable (big differences in value and error). If sample 9 were considered an outlier and suppressed in Table 1, the resulting average, about 0.50 would not be much affected.
Answer:
We are aware of this work and we consider it would be useful to modify this method of interpreting primary viscometric data to more complex systems/situations presented in our work. An addition to the text is made on page 7.
Regarding the convergence of [eta] measurements obtained by different operators. Sometimes we carry out such testing starting from one dry sample. Such a test was carried out, for example, for one sample of alkylated copolymer N-methyl-N-vinylacetamide and N-methyl-N-vinylamine (MVAA-co-МVACnH2n+1) in 0.1M. As a result of two independent measurements, the following average values were obtained: [eta]=(74±3)cm3/g and k’’= +(0.25+-0.07), which are 4 and 28%. The uncertainty is greater when estimating the dimensionless parameter. As to fraction 11 in Table 1. Indeed, this is a fallout from the general range of values of dimensionless parameters. However, the main quantity is [eta] and it is within reasonable limits in the series of molecular weights. In principle, such a result requires a second measurement, but in this case there is simply no object left for research (sometimes this happens, especially with the last fractions).
- 4-5-6 The finding that for a conventional non-associating, non-electrolyte polymer in good solvents, k’-k” = 0.50+-0.015 (+-3%) with such small variation for over 12 samples is really rewarding. Good!! However, in section 3.1.2, the results for hydrophobic HPMC, k’-k” = 0.70 and k’-k” = 1.22 for the amphiphilic MV copolymer are both for only one viscometric experiment (just one sample)
Answer: Concerning HPMC. This cellulose derivative is soluble in water and, strictly speaking, cannot be called hydrophobic. In this regard, we have changed the title of the section 3.1.2 to Peculiarities of Viscosity Behavior of Associating Systems. We have carried out viscometric studies of one sample of HPMC. Unfortunately, the archive of our earlier and wider study of HPMC samples in 2002-2004 is no longer available. However, it can be stated from those results that the k’ value for HPMC in water is 1.1+-0.2. This is an indication of the HPMC's tendency to self-associate. We are going to continue the studies.
- In the second part of the paper, the authors present quite original treatments of the experimental data. The one based on ln eta_r vs. c[eta] (section 2.2). is quite simple and effective- Fig. 5 provides a nice classification of the behaviors of different (solvent affinity) of the polymer systems .As B_2=k’’, it would be wise to give the B_2 values for the parabolic curves in the legends of Figures 5 and 6 or in the text. The other treatment based on the lg([eta]M_L vs. lg L plot (section 2.3), proposed by Pavlov’s group in refs. 71 and 72. It is a bit speculative but seemingly correct,, and deserves to be considered by workers in this field. I just don't see the values (or references) for the M_L of the various polymers.
Answer: We added a column with B2 values to Table 4 and in the legends of Fig. 5. The ML values for the considered systems are given in the legend to Fig.9
- Finally, some text editing:
- 2 “As de la Torre et al…” Please, write “As Garcia de la Torre et al…”
Answer: is corrected
- 5, Figure 1, legend of Y axis, I think ln eta_r should be ln eta_r/c
Answer: is corrected
- 10 Figure 5: broken text: [S. G. Weissberg,]
Answer:is removed.
- 4 “The polymers were… Although it is later clear, please say right here that
they are in water or NaCl aqueous solution
Answer: corrections are given
- 4 “…as Flory noted, extrapolation (lnηr/c)c→0 is more preferable… Please cite the Flory’s publication were this was affirmed. Is it ref. 39 or ref. 66?
Answer: The reference was indicated as [66], we added the exact page number - 325: Flory, P.J. Principles of polymer chemistry; Cornell university press: 1953. P. 325

Reviewer 2 Report
Thank you very much I give my opinion on this article which is titled:
New facet in viscometry of charged associating polymer sys- 2
time in dilute solutions
the work is old, that is to say is not current and outdated. I see that there are many scientific mistakes that are not clear and the figures are not well presented and well written AND also are parachute I see that this work is not good and not be accepted thank you.
Author Response
No comments.
Reviewer 3 Report
The authors have examined on " New facet in viscometry of charged associating polymers systems in dilute solutions". It is interesting study for fundamental information. Nevertheless, I would ask the author if possible they could work on the Figures and Figure legends representation. The manuscript is publishable.
Author Response
Dear Reviewer,
Thank you very much for your careful reading of our work and useful comments.
We have tried to present the plots and captions to them in an improved form.